# Federated Graph Learning with Graphless Clients

**Xingbo Fu**                                                                 *xf3av@virginia.edu*
*Department of Electrical and Computer Engineering*
*University of Virginia*

**Song Wang**                                                                 *sw3wv@virginia.edu*
*Department of Electrical and Computer Engineering*
*University of Virginia*

**Yushun Dong**                                                               *yd6eb@virginia.edu*
*Department of Electrical and Computer Engineering*
*University of Virginia*

**Binchi Zhang**                                                              *epb6gw@virginia.edu*
*Department of Electrical and Computer Engineering*
*University of Virginia*

**Chen Chen**                                                                 *zrh6du@virginia.edu*
*Department of Computer Science*
*University of Virginia*

**Jundong Li**                                                                *jundong@virginia.edu*
*Department of Electrical and Computer Engineering*
*University of Virginia*

**Reviewed on OpenReview:** *https://openreview.net/forum?id=mVApOeDfyR*

## Abstract

Federated Graph Learning (FGL) is tasked with training machine learning models, such as Graph Neural Networks (GNNs), for multiple clients, each with its own graph data. Existing methods usually assume that each client has both node features and graph structure of its graph data. In real-world scenarios, however, there exist federated systems where only a part of the clients have such data while other clients (i.e. *graphless clients*) may only have node features. This naturally leads to a novel problem in FGL: *how to jointly train a model over distributed graph data with graphless clients?* In this paper, we propose a novel framework FedGLS to tackle the problem in FGL with graphless clients. In FedGLS, we devise a local graph learner on each graphless client which learns the local graph structure with the structure knowledge transferred from other clients. To enable structure knowledge transfer, we design a GNN model and a feature encoder on each client. During local training, the feature encoder retains the local graph structure knowledge together with the GNN model via knowledge distillation, and the structure knowledge is transferred among clients in global update. Our extensive experiments demonstrate the superiority of the proposed FedGLS over five baselines.

## 1 Introduction

Recent years have witnessed a growing development of graph-based applications in a wide range of high-impact domains. As a powerful deep learning tool for graph-based applications, Graph Neural Networks (GNNs) exploit abundant information inherent in graphs (Wu et al., 2020) and show superior performance in different domains, such as node classification (Fu et al., 2023; He et al., 2022) and link prediction (Tan

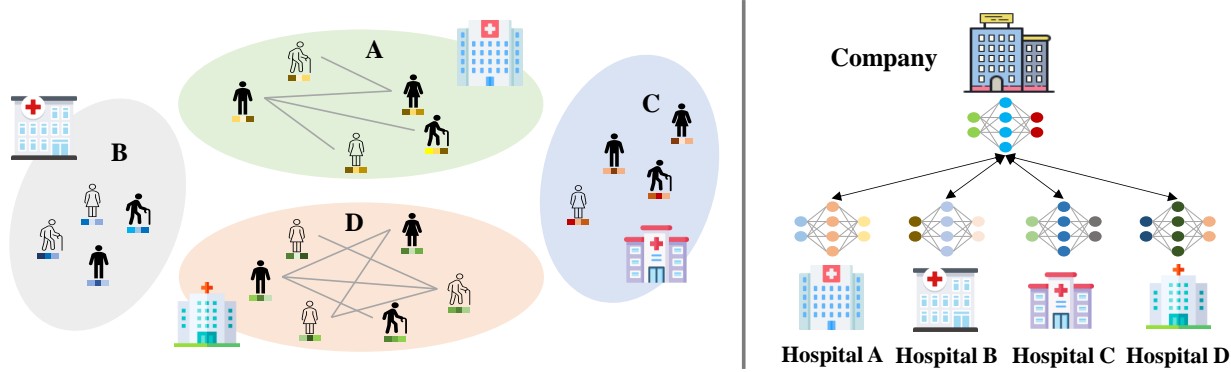

Figure 1: An example of a healthcare system including four hospitals. In this example, Hospital A and Hospital D have their local datasets of patients (node features) and co-staying information (links) among them. In the meantime, Hospital B and Hospital C only have their local datasets of patients (node features). The four hospitals aim to jointly train a model for predicting whether a patient is at high risk of contracting a contagious disease, orchestrated by a third-party company over their local datasets while the company cannot directly access their private datasets.

et al., 2023a; Zhang & Chen, 2018). Traditionally, GNNs are trained over graph data stored on a single machine. In the real world, however, multiple data owners (i.e., clients) may have their own graph data and hope to jointly train GNNs over their graph data. The challenge in this scenario is that the graph data is often not allowed to be collected from different places to one single server for training due to the emphasis on data security and user privacy (Voigt & Von dem Bussche, 2017; Wang et al., 2024b). Considering a group of hospitals, for instance, each of them has its local dataset of patients. These hospitals hope to collaboratively train GNNs for patient classification tasks (e.g., predicting whether a patient is at high risk of contracting a contagious disease) while keeping their private patient data locally due to strict privacy policies and commercial competition.

To tackle the above challenge, Federated Learning (FL) (McMahan et al., 2017) enables collaborative training among clients over their private graph data orchestrated by a central server. Typically, FL can be categorized into horizontal and vertical FL based on how data is distributed among clients (Yang et al., 2019). This study focuses on horizontal FL where distributed datasets share the same feature space. During each training round, the selected clients receive the global model from the central server and perform local updates over their local graph data. The central server aggregates the updated local models from the clients and computes the new global model for the next training round. Numerous studies have been proposed to improve FL performance over graph data by reconstructing cross-client information (Zhang et al., 2021b;a), aligning overlapping instances (Peng et al., 2021; Zhou et al., 2022), and mitigating data heterogeneity (Xie et al., 2021; Tan et al., 2023b; Fu et al., 2024).

The above methods rely on a fundamental assumption that each client has graph structure information of its local graph data. In the real world, however, this assumption may not be feasible for all the clients. Instead, there may be a part of clients only having local node features, whereas other clients have both node features and edge information. To illustrate this scenario in practice, we provide a real-world example as follows. More practical examples can be found in Appendix A.

**Motivating example.** Considering the aforementioned example of a healthcare system as shown in Figure 1, we may construct local graphs in each hospital by taking patient demographics as node features and co-staying in a ward as edges. In the real world, however, some hospitals may not record the co-staying information and cannot construct patient graphs. As a result, these hospitals are unable to directly train GNNs to predict whether a patient is at high risk of contracting a contagious disease in a federated manner. If we instead train a machine learning model only based on patient features, the classification performance will

be unsatisfactory because we disregard the important co-staying information between patients, which can significantly determine the risk of contracting a contagious disease for a patient.

The above scenario brings us a novel problem in the federated setting: *how to jointly train a GNN model for the classification task from isolated graphs distributed in multiple clients while some clients only have node features?* In this paper, we name such clients as *graphless clients*. Since directly training GNNs is obviously infeasible in this setting, collaboratively training non-GNN-based models such as multi-layer perceptrons (MLPs) and Support Vector Machines (SVMs) is a plausible solution to the above problem. However, a number of experiments in prior works have demonstrated that the non-GNN-based models are typically less accurate than GNNs for the classification task (Franceschi et al., 2019; Zhang et al., 2022). Another intuitive method is to let a graphless client construct graph structure based on the similarity of the features (e.g., using kNN (Gidaris & Komodakis, 2019)) and jointly train GNNs together with other clients. A disadvantage of this method is that the generated graphs are only dependent on node features and are not suitable for node classification (Franceschi et al., 2019; Liu et al., 2022). To overcome the disadvantage, it is natural to let the graphless clients produce graph structures with the structure knowledge of other clients. However, structure knowledge transfer and utilization in a federated manner are still challenging and unexplored.

In this study, we propose a novel framework FedGLS to handle FGL with graphless clients. FedGLS aims to solve two key challenges of utilizing structure knowledge in this scenario: 1) how to transfer structure knowledge among clients; and 2) how to utilize the transferred knowledge on graphless clients? In FedGLS, we first design two modules - a GNN model and a feature encoder on each client. In particular, we deploy a third module - a local graph learner on each graphless client. The GNN model learns node embeddings over the local graph and the feature encoder approximates the output of the GNN model via knowledge distillation (Hinton et al., 2015). Therefore, the GNN model and the feature encoder together retain structure knowledge on each client. The central server collects the local parameters of the two modules from the clients and gets the global parameters. In this way, FedGLS transfers structure knowledge among clients. The local graph learner utilizes the structure knowledge by maximizing the consistency between the output of the global GNN model and feature encoder on each graphless client with a contrastive loss. We conduct extensive experiments over five datasets, and the results show that FedGLS outperforms other baselines.

Our contributions can be summarized as follows.

- **Problem Formulation.** We propose a novel research problem of FGL with graphless clients and provide the formal definition of the proposed problem.

- **Algorithm Design.** We propose FedGLS to tackle the proposed problem. We devise a scheme for transferring structure knowledge among clients in FedGLS by letting a feature encoder imitate the node embeddings from a GNN model. The scheme enables a graph learner on each graphless client to learn its local graph structure with the structure knowledge transferred from other clients.

- **Experimental Evaluations.** We conduct extensive experiments on real-world datasets, and the results validate the superiority of our proposed FedGLS against five baselines. Our implementation of FedGLS is available in the supplementary materials.

## 2 Related Work

### 2.1 Federated Learning

FL (McMahan et al., 2017) enables participants (i.e., clients) to jointly train a model under the coordination of a central server without sharing their private data. One key problem that FL concerns is statistical heterogeneity: the data across clients are likely to be non-IID (Li et al., 2020a; Wu et al., 2024a). When each client updates its local model based on its local dataset, its local objective may be far from the global objective. Thus, the averaged global model is away from the global optima (Wang et al., 2024a; Wu et al., 2024b; Wang et al., 2023) and influences the convergence of FL. To overcome the performance degradation of FedAvg when data at each client are statistically heterogeneous (non-IID), a number of recent studies have been proposed from different aspects. Typically, these studies can be categorized into single global

model-based methods and personalized model-based methods. Single global model-based methods aim to train a global model for all clients. For instance, FedProx (Li et al., 2020b) adds a proximal term to the local training loss to keep updated parameters close to the global model. SCAFFOLD (Karimireddy et al., 2020) customizes the gradient updates of personalized models to mitigate client drifts between local models and the global model. Moon (Li et al., 2021) uses a contrastive loss to increase the distance between the current and previous local models. Personalized model-based methods instead enable each client to train a personalized model to mitigate the impact of data heterogeneity. For example, pFedHN (Shamsian et al., 2021) trains a central hypernetwork to output a unique personal model for each client. FedProto (Tan et al., 2022) and FedProc (Mu et al., 2023) utilize the prototypes to regularize the local model training.

## 2.2 Federated Graph Learning

Following many well-designed FL methods for Euclidean data (e.g., images), a number of recent studies have begun tackling challenges in FL on graph data (Fu et al., 2022) to achieve better performance on downstream tasks. One specific problem in FL on graph data is missing neighboring information when each client only owns a part of the original graph. The recent proposed methods recover missing neighboring information by transmitting intermediate result (Zhang et al., 2021a) and generating missing neighbors (Zhang et al., 2021b). Another interesting issue in FL on graphs is aligning overlapping instances across clients. This issue happens in FL with heterogeneous graphs (Peng et al., 2021) and vertical FL on graphs (Zhou et al., 2022). In the meantime, some recent studies focus on the unique challenges caused by data heterogeneity in FGL. For example, GCFL (Xie et al., 2021) enables clients with similar graph structure properties to share model parameters. FedStar (Tan et al., 2023b) designs a structure encoder to share structure knowledge among clients for graph classification. FedLit (Xie et al., 2023) mitigates the impact of link-type heterogeneity underlying homogeneous graphs in FGL via an EM-based clustering algorithm. Different from the aforementioned problems where the clients own structured data (i.e., graphs), our work aims to deal with the setting where a part of the clients do not have structure information.

## 3 Problem Statement

In this section, we first present basic concepts in GNNs and FL. Then we propose a novel problem setting of FGL with graphless clients.

### 3.1 Preliminaries

Before formally presenting the formulation of the novel problem, we first introduce the concepts of GNNs and FGL.

**Notations.** We use bold uppercase letters (e.g., $\mathbf{A}$) to represent matrices. For any matrix, e.g., $\mathbf{Z}$, we use the corresponding bold lowercase letters $\mathbf{z}_i$ to denote its $i$-th row vector. We use letters in calligraphy font (e.g., $\mathcal{V}$) to denote sets. $|\mathcal{V}|$ denotes the cardinality of set $\mathcal{V}$.

**Graph Neural Networks.** We use $\mathcal{G} = (\mathcal{V}, \mathcal{E}, \mathbf{X})$ to denote an attributed graph, where $\mathcal{V} = \{v_1, v_2, \cdots, v_n\}$ is the set of $n = |\mathcal{V}|$ nodes, $\mathcal{E}$ is the edge set, and $\mathbf{X} \in \mathbb{R}^{n \times d}$ is the node feature matrix. $d$ is the number of node features. The edges describe the relations between nodes and can also be represented by an adjacency matrix $\mathbf{A} \in \mathbb{R}^{n \times n}$. A GNN model $f$ parameterized by $\theta = \{\theta_z, \theta_c\}$ learns the node embeddings $\mathbf{Z} \in \mathbb{R}^{n \times d'}$ based on $\mathbf{X}$ and $\mathbf{A}$

$$\mathbf{Z} = f(\mathbf{X}, \mathbf{A}; \theta_z). \tag{1}$$

Here $d'$ is the embedding dimension, and $\theta_z$ represents parameters of the encoder part in $f$ to obtain the node embedding $\mathbf{Z}$. For the node classification task, we use $\mathbf{Z}$ to obtain the prediction $\hat{\mathbf{Y}} = f(\mathbf{Z}; \theta_c) \in \mathbb{R}^{n \times p}$ where $p$ is the number of classes, and $\theta_c$ represents parameters of the predictor in $f$. Given the label set $\mathcal{Y}_L = \{y_1, y_2, \cdots, y_L\}$ where $y_i$ denotes the label of node $v_i \in \mathcal{V}_L = \{v_1, v_2, \cdots, v_L\}$, the objective is to minimize the difference between $\hat{\mathbf{Y}}$ and $\mathcal{Y}_L$

$$\min_{\theta} \mathcal{L}(\theta) = \frac{1}{|\mathcal{V}_L|} \sum_{v_i \in \mathcal{V}_L} \ell(\hat{\mathbf{y}}_i, y_i), \tag{2}$$

where $\ell(\cdot, \cdot)$ is the cross-entropy loss.

**Federated Graph Learning.** In a federated system with $K$ clients $\mathcal{C} = \{c^{(1)}, c^{(2)}, \cdots, c^{(K)}\}$, each client $c^{(k)} \in \mathcal{C}$ owns a private graph $\mathcal{G}^{(k)} = (\mathcal{V}^{(k)}, \mathcal{E}^{(k)}, \mathbf{X}^{(k)})$ and $n^{(k)} = |\mathcal{V}^{(k)}|$. The goal of the clients is to collaboratively train a GNN model $f$ parameterized by $\theta$ orchestrated by a central server while keeping the private datasets locally. Specifically, the objective is to solve

$$\arg\min_{\theta} \mathcal{F}(\theta) := \sum_{k=1}^{K} \frac{n^{(k)}}{N} \mathcal{L}^{(k)}(\theta), \tag{3}$$

where $\mathcal{L}^{(k)}(\theta) = \frac{1}{|\mathcal{V}_L^{(k)}|} \sum_{v_i \in \mathcal{V}_L^{(k)}} \ell(\hat{\mathbf{y}}_i, y_i)$, and $N = \sum_{k=1}^{K} n^{(k)}$ is the total number of nodes around all the clients. FedAvg (McMahan et al., 2017) is one of generic federated optimization methods, which can be directly applied to FL on graph data. Typically, during each training round, a client $c^{(k)}$ updates its local GNN parameters $\theta^{(k)}$ over its private graph $\mathcal{G}^{(k)}$ via SGD for a number of epochs with initialization of the parameters set to the global GNN parameters $\theta$. At the end of the round, the server collects $\{\theta^{(k)}\}_{k=1}^{K}$ from clients and computes the new global GNN parameters by

$$\theta = \sum_{k=1}^{K} \frac{n^{(k)}}{N} \theta^{(k)}. \tag{4}$$

The new global GNN parameters $\theta$ are used for local training in the next round.

### 3.2 Problem Definition

The clients jointly training a GNN model require structure information (e.g., $\mathbf{A}^{(k)}$) of a private graph $\mathcal{G}^{(k)}$ known by each client $c^{(k)}$. However, this requirement may not be feasible for all the clients and some of the clients only have its local node feature matrix $\mathbf{X}^{(k)}$. As a result, we cannot obtain a GNN model trained directly over the distributed graph data on all the clients.

Based on the aforementioned challenges, we propose a novel problem setting in FGL. We provide a formal definition of the problem as follows.

PROBLEM 1. **Federated graph learning with graphless clients**: *Given a set of $K$ clients $\mathcal{C} = \{c^{(k)}\}_{k=1}^{K}$ and $1 < M < K$, each client $c^{(k)} \in \mathcal{C}_1 = \{c^{(k)}\}_{k=1}^{M}$ owns the node features $\mathbf{X}^{(k)}$ in its private graph $\mathcal{G}^{(k)}$ with the complete structure information (e.g., $\mathbf{A}^{(k)}$) while each graphless client $c^{(k)} \in \mathcal{C}_2 = \{c^{(k)}\}_{k=M+1}^{K}$ only owns its local node features $\mathbf{X}^{(k)}$. The goal of the whole clients $\mathcal{C}$ is to collaboratively train a model for the node classification task without sharing their graph data.*

## 4 Methodology

In this section, we present the proposed framework FedGLS. The goal of FedGLS is to let graphless clients learn local graph structures with the structure knowledge transferred from other clients. To achieve this goal, FedGLS solves the following two challenges: 1) how to transfer structure knowledge among clients; and 2) how to utilize the transferred structure knowledge on graphless clients. To handle the first challenge, we design a GNN model and a feature encoder on each client. The feature encoder aims to retain structure knowledge together with the GNN model via knowledge distillation. To utilize the transferred structure knowledge, we design a graph learner on each graphless client. This module generates local graph structure and learns the structure knowledge in the GNN model and the feature encoder via a contrastive loss.

### 4.1 Framework Overview

Figure 2 illustrates an overview of FedGLS. It consists of two training stages: local training on the clients and global update on the central server.

**Local Training.** On each client, a GNN model generates node embeddings with respect to node features and local graph structure. The goal of the feature encoder is to retain structure knowledge together with the

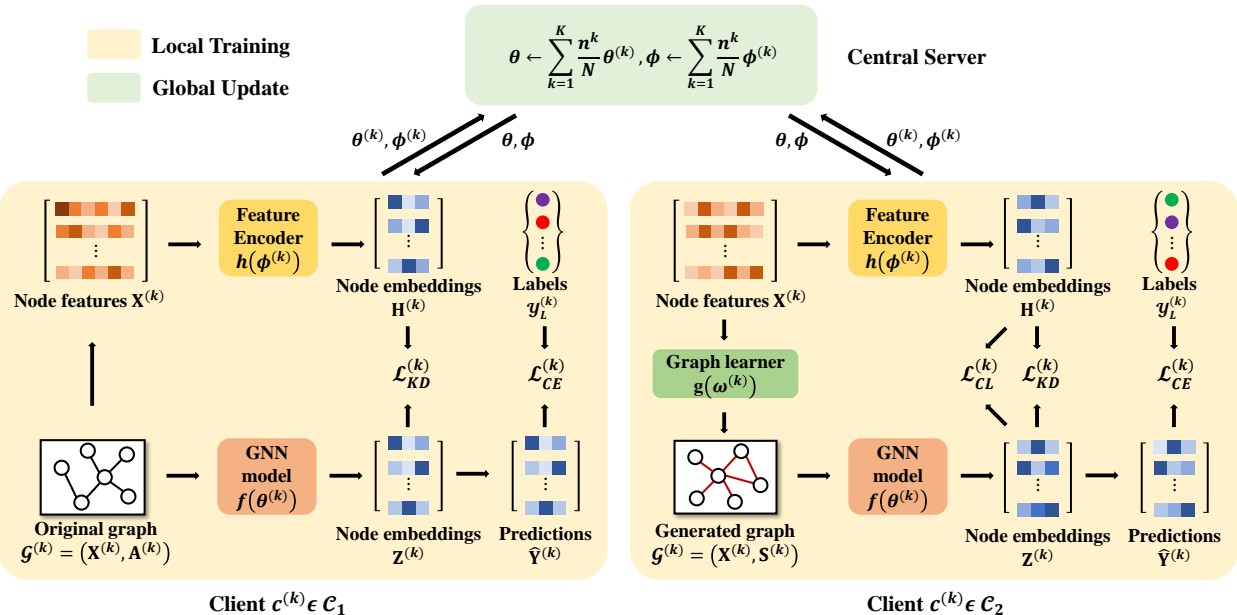

Figure 2: An overview of the proposed FedGLS.

GNN model. It approximates the output (i.e., node embeddings) of the GNN model using the knowledge learned by the GNN model. On each graphless client, a graph learner generates local graph structure and learns the structure knowledge via a contrastive loss. Finally, the well-trained graph learner produces local graph structure and the GNN model learns more expressive node embeddings.

**Global Update.** After local training, the central server gathers the local parameters of the GNN model and the feature encoder from the clients. Then it computes the new global parameters following FedAvg and broadcasts them to the clients for local training in the next round.

## 4.2 Local Training

As introduced above, we deploy a GNN model and a feature encoder on each client. For each graphless client, an extra graph learner is deployed to generate the graph structure. In this subsection, we will introduce the local update of the three modules during local training on the clients.

For each graphless client $c^{(k)} \in \mathcal{C}_2$, a graph learner $g$ produces an adjacency matrix $\mathbf{S}^{(k)} \in \mathbb{R}^{n^{(k)} \times n^{(k)}}$ with respect to the node feature matrix $\mathbf{X}^{(k)}$. Formally, we formulate the graph learner as

$$\mathbf{S}^{(k)} = g(\mathbf{X}^{(k)}; \omega^{(k)}) = \Omega(\text{Gen}(\mathbf{X}^{(k)}; \omega^{(k)})), \tag{5}$$

where $\omega^{(k)}$ denotes the parameters in $g$. $\text{Gen}(\cdot)$ denotes a graph generator which produces a matrix $\tilde{\mathbf{S}}^{(k)} \in \mathbb{R}^{n^{(k)} \times n^{(k)}}$ based on the node features. Typically, we instantiate the graph generator as an MLP encoder or an attentive encoder followed by a cosine similarity function. $\Omega(\cdot)$ is a non-parametric adjacency processor which conducts post-processing on $\tilde{\mathbf{S}}^{(k)}$ and outputs $\mathbf{S}^{(k)}$. Generally, it includes four main operations: *sparsification*, *activation*, *symmetrization*, and *normalization* (Liu et al., 2022). The adjacency processor $\Omega(\cdot)$ ensures that $\mathbf{S}^{(k)}$ is a normalized symmetric sparse adjacency matrix with non-negative values. For detailed descriptions about designing graph generators and adjacency processors in the graph learner, readers can refer to Appendix B.

Then the GNN model $f$ on each graphless client $c^{(k)} \in \mathcal{C}_2$ produces node embeddings with the generated adjacency matrix $\mathbf{S}^{(k)}$ and the node feature matrix $\mathbf{X}^{(k)}$ as the input. For each client $c^{(k)} \in \mathcal{C}_1$, the client

instead uses the original adjacency matrix $\mathbf{A}^{(k)}$. Specifically, the GNN model produces the node embeddings $\mathbf{Z}^{(k)}$ by

$$\mathbf{Z}^{(k)} = f(\mathbf{X}^{(k)}, \mathbf{S}^{(k)}; \theta^{(k)}) \tag{6}$$

for each client $c^{(k)} \in \mathcal{C}_2$; otherwise, $\mathbf{S}^{(k)}$ will be replaced by the original adjacency matrix $\mathbf{A}^{(k)}$. $\theta^{(k)}$ is the parameters in $f$. In this paper, we instantiate the GNN model $f$ as a GCN.

In the meantime, the feature encoder $h$ similarly generates the node representations $\mathbf{H}^{(k)}$ but it is only based on the node feature matrix $\mathbf{X}^{(k)}$

$$\mathbf{H}^{(k)} = h(\mathbf{X}^{(k)}; \phi^{(k)}), \tag{7}$$

where $\phi^{(k)}$ denotes the parameters in $h$. In this paper, we instantiate the feature encoder $h$ as an MLP.

**Optimizing $\omega^{(k)}$.** For a graphless client $c^{(k)} \in \mathcal{C}_2$, its well-trained graph learner $g$ with its parameters $\omega^{(k)}$ is supposed to learn structure knowledge transferred from other clients. Typically, it produces the adjacency matrix $\mathbf{S}^{(k)}$ so that the node embeddings $\mathbf{Z}^{(k)}$ produced by the GNN model based on $\mathbf{S}^{(k)}$ and $\mathbf{X}^{(k)}$ are consistent with $\mathbf{H}^{(k)}$ produced by the feature encoder. To achieve this, we optimize the local graph learner by maximizing the agreement with a contrastive loss (e.g., NT-Xent (Chen et al., 2020a)). Specifically, we consider the embedding pair $\mathbf{z}_i^{(k)}$ and $\mathbf{h}_i^{(k)}$ of node $v_i \in \mathcal{V}^{(k)}$ as a positive pair. In contrast, the embedding $\mathbf{z}_i^{(k)}$ and the embedding of any other node $v_j \in \mathcal{V}^{(k)}$ (either in $\mathbf{Z}^{(k)}$ or $\mathbf{H}^{(k)}$) compose a negative pair. Our goal is to decrease the distance between positive pairs and increase the distance between negative pairs. Concretely, we formalize it as the contrastive loss as follows:

$$\ell_i^{(k)}(\mathbf{Z}^{(k)}, \mathbf{H}^{(k)}) = -\log \frac{e^{\text{sim}(\mathbf{z}_i^{(k)}, \mathbf{h}_i^{(k)})/\tau}}{\sum_{j=1}^{n^{(k)}} \mathbb{1}_{[i \neq j]} \big[ e^{\text{sim}(\mathbf{z}_i^{(k)}, \mathbf{h}_j^{(k)})/\tau} + e^{\text{sim}(\mathbf{z}_i^{(k)}, \mathbf{z}_j^{(k)})/\tau} \big]}, \tag{8}$$

where $\text{sim}(\cdot, \cdot)$ is the cosine similarity function, and $\tau$ is a temperature parameter. The contrastive loss encourages $\mathbf{z}_i^{(k)}$ not to be too close to other node embeddings and therefore alleviate the over-smoothing issue in GNNs.

In the end, the local graph learner is optimized by minimizing the total contrastive loss:

$$\min_{\omega^{(k)}} \mathcal{L}_{CL}^{(k)} = \frac{1}{n^{(k)}} \sum_{i=1}^{n^{(k)}} \ell_i(\mathbf{Z}^{(k)}, \mathbf{H}^{(k)}). \tag{9}$$

**Optimizing $\theta^{(k)}$.** The GNN model $f$ aims to learn expressive node embeddings and therefore predicts the labels of unlabeled nodes. To optimize the parameters $\theta^{(k)}$ in $f$ on each client $c^{(k)} \in \mathcal{C}$, we minimize the classification loss over all labeled nodes in $\mathcal{V}_L^{(k)}$ by

$$\min_{\theta^{(k)}} \mathcal{L}_{CE}^{(k)} = \frac{1}{|\mathcal{V}_L^{(k)}|} \sum_{v_i \in \mathcal{V}_L^{(k)}} \text{CE}(\hat{\mathbf{y}}_i, y_i), \tag{10}$$

where $\text{CE}(\cdot, \cdot)$ denotes the cross-entropy loss.

**Optimizing $\phi^{(k)}$.** The goal of the feature encoder is to produce $\mathbf{H}^{(k)}$ without structure information which is consistent with $\mathbf{Z}^{(k)}$ and therefore enables the training of the graph learner. The key challenge for training graph learners is to enforce closeness between $\mathbf{H}^{(k)}$ and $\mathbf{Z}^{(k)}$. To tackle this issue, we resort to knowledge distillation (Hinton et al., 2015). The intuition of knowledge distillation is to let a student model learn with the knowledge (e.g., predictions) from a teacher model. Typically, the student model is able to produce comparable outputs with the teacher model. In FedGLS, we choose to distill knowledge from the GNN model to the feature encoder. Then the feature encoder (e.g., an MLP) achieves comparable performance with a GNN via learning the knowledge transferred from the GNN only based on the features (Zhang et al., 2022). In FedGLS, the parameters $\phi^{(k)}$ of a feature encoder on each client $c^{(k)} \in \mathcal{C}$ are updated by approximating the knowledge (i.e., the node embeddings $\mathbf{Z}^{(k)}$) from its GNN model. Specifically, the feature encoder on

client $c^{(k)} \in \mathcal{C}$ is optimized by minimizing the discrepancy between $\mathbf{h}_i^{(k)}$ and $\mathbf{z}_i^{(k)}$ for $v_i \in \mathcal{V}^{(k)}$ via knowledge distillation (Hinton et al., 2015) as follows:

$$\min_{\phi^{(k)}} \mathcal{L}_{KD}^{(k)} = \sum_{v_i \in \mathcal{V}^{(k)}} \mathrm{KL}(f(\mathbf{z}_i^{(k)}; \theta_c^{(k)}) || f(\mathbf{h}_i^{(k)}; \theta_c^{(k)})), \tag{11}$$

where $\mathrm{KL}(\cdot || \cdot)$ is to compute the Kullback-Leibler divergence (KL-divergence).

### 4.3 Global Update

During global update, the central server gathers the local $\theta^{(k)}$ and $\phi^{(k)}$ from the clients. Then it computes the new global $\theta$ and $\phi$ with FedAvg. More specifically, the new global $\theta$ and $\phi$ for the next round are calculated by

$$(\theta, \phi) \leftarrow \Big( \sum_{c^{(k)} \in \mathcal{C}} \frac{n^{(k)}}{N} \theta^{(k)}, \sum_{c^{(k)} \in \mathcal{C}} \frac{n^{(k)}}{N} \phi^{(k)} \Big). \tag{12}$$

### 4.4 Overall Algorithm

The overall federated training algorithm of FedGLS is shown in Algorithm 1. During each round, the central server sends global $\theta$ and $\phi$ to the selected clients. For each client $c^{(k)} \in \mathcal{C}_2$, the graph learner $g$ first produces the adjacency matrix $\mathbf{S}^{(k)}$. The GNN model $f$ takes node features $\mathbf{X}^{(k)}$ and the generated adjacency matrix $\mathbf{S}^{(k)}$ (for clients in $\mathcal{C}_1$, they use their original adjacency matrices $\mathbf{A}^{(k)}$ instead) to get node representations $\mathbf{Z}^{(k)}$. Then the feature encoder $h$ computes corresponding node embeddings $\mathbf{H}^{(k)}$ only with node features $\mathbf{X}^{(k)}$ as input. The parameters $\omega^{(k)}$ in $g$ are updated by minimizing the discrepancy between $\mathbf{Z}^{(k)}$ and $\mathbf{H}^{(k)}$ using Eq. (9). Afterward, each client $c^{(k)} \in \mathcal{C}$ updates the parameters $\theta^{(k)}$ of $f$ via supervised learning using Eq. (10) and updates the parameters $\phi^{(k)}$ of $h$ with the knowledge distilled from the GNN model as supervision information using Eq. (11). Finally, the central server collects the updated $\theta^{(k)}$ and $\phi^{(k)}$ from the clients to get the new global $\theta$ and $\phi$ using Eq. (12) for local training in the next round. We provide complexity analysis Appendix C.

## 5 Experiments

We conduct extensive experiments over five real-world datasets to verify the superiority of the proposed FedGLS. In particular, we aim to answer the following questions.

**RQ1:** How does FedGLS perform compared with other state-of-the-art baselines?

**RQ2:** How well can FedGLS be stable under different local epochs and various graphless client ratios?

### 5.1 Experiment Setup

#### 5.1.1 Datasets

We synthesize the distributed graph data based on five common real-world datasets, i.e., Cora (Sen et al., 2008), CiteSeer (Sen et al., 2008), PubMed (Sen et al., 2008), Flickr (Zeng et al., 2020), and ogbn-arxiv (Hu et al., 2020). Following the data partition strategy in previous studies (Huang et al., 2023; Zhang et al., 2021b), we synthesize the distributed graph data by splitting each dataset into multiple communities via the Louvain algorithm (Blondel et al., 2008); each community is regarded as an entire graph in a client. We summarize the statistics and basic information about the datasets in Appendix D.

In our experiments, we randomly select half clients as graphless clients. Following the setting in (Zhang et al., 2021b), we randomly select nodes on each client and let 60% for training, 20% for validation, and the remaining for testing. We report the average accuracy for node classification over the clients for five random repetitions.

---

**Algorithm 1** The detailed algorithm of FedGLS.

---

**Input**: global parameters $\theta, \phi$; learning rate $\alpha, \beta, \gamma$, local epoch $E$
**Output**: $\theta$

1: **repeat**
2:     Server selects a subset of clients $\mathcal{C}_s$ from $\mathcal{C}$, then broadcasts $\theta$ and $\phi$ to $\mathcal{C}_s$
3:     **for** client $c^{(k)} \in \mathcal{C}_s$ **do**
4:         $\theta^{(k)} \leftarrow \theta$, $\phi^{(k)} \leftarrow \phi$
5:         **if** $c^{(k)} \in \mathcal{C}_2$ **then**
6:             // Update graph learner
7:             Compute $\mathcal{L}_{CL}^{(k)}(\omega^{(k)}) = \frac{1}{n^{(k)}} \sum_{i=1}^{n^{(k)}} \ell_i(\mathbf{Z}^{(k)}, \mathbf{H}^{(k)})$
8:             $\omega^{(k)} \leftarrow \omega^{(k)} - \gamma \nabla_{\omega^{(k)}} \mathcal{L}_{CL}^{(k)}(\omega^{(k)})$
9:         **end if**
10:         **for** $i = 1, 2 \cdots, E$ **do**
11:             // Update GNN classifier
12:             Compute $\mathcal{L}_{CE}^{(k)}(\theta^{(k)}) = \frac{1}{|\mathcal{V}_L^{(k)}|} \sum_{v_i \in \mathcal{V}_L^{(k)}} \mathrm{CE}(\hat{\mathbf{y}}_i, y_i)$
13:             $\theta^{(k)} \leftarrow \theta^{(k)} - \alpha \nabla_{\theta^{(k)}} \mathcal{L}_{CE}^{(k)}(\theta^{(k)})$
14:             // Update feature encoder
15:             Compute $\mathcal{L}_{KD}^{(k)}(\phi^{(k)}) = \sum_{v_i \in \mathcal{V}^{(k)}} \mathrm{KL}(f(\mathbf{z}_i^{(k)}; \theta_c^{(k)}) || f(\mathbf{h}_i^{(k)}; \theta_c^{(k)}))$
16:             $\phi^{(k)} \leftarrow \phi^{(k)} - \beta \nabla_{\phi^{(k)}} \mathcal{L}_{KD}^{(k)}(\phi^{(k)})$
17:         **end for**
18:         Client $c^{(k)}$ sends $\theta^{(k)}, \phi^{(k)}$ back to server
19:     **end for**
20:     Server updates $\theta$ and $\phi$ via $\theta \leftarrow \sum_{c^{(k)} \in \mathcal{C}} \frac{n^{(k)}}{N} \theta^{(k)}, \phi \leftarrow \sum_{c^{(k)} \in \mathcal{C}} \frac{n^{(k)}}{N} \phi^{(k)}$
21: **until** training stop

---

### 5.1.2 Baselines

Since FedGLS is proposed to deal with a novel problem setting in FGL with graphless clients, most of the existing frameworks cannot be directly adopted without any preprocessing. Considering this, we first design the following two baselines.

- **Fed-MLP**: the clients in $\mathcal{C}$ jointly train an MLP model;

- **Fed-GNNMLP**: the clients in $\mathcal{C}_1$ collaboratively train a GNN model while the clients in $\mathcal{C}_2$ collaboratively train an MLP model;

In the meantime, we include the following baseline which can be adopted to handle the heterogeneous model architecture setting.

- **FedProto** (Tan et al., 2022): the local models on the clients in $\mathcal{C}_1$ are GNNs and those on the clients in $\mathcal{C}_2$ are MLPs, then FedProto performs collaborative training by aggregating prototypes. A prototype is the mean value of the embeddings of instances in a class.

Furthermore, we also consider the following two baselines with preprocessing.

- **Local-GNNk**: it first performs the kNN operation on the graphless clients in $\mathcal{C}_2$, then each client in $\mathcal{C}$ individually trains a GNN model over its local data without any communication among the clients;

- **Fed-GNNk**: it first performs the kNN operation on the graphless clients in $\mathcal{C}_2$, then jointly trains a GNN model over the clients in $\mathcal{C}$.

Table 1: Node classification performance (Accuracy± Std) over different datasets. Bold and underlined values indicate best and second-best mean performances, respectively.

| Datasets | Fed-MLP | Fed-GNNMLP | FedProto | Local-GNNk | Fed-GNNk | FedGLS | Fed-GNN |
|---|---|---|---|---|---|---|---|
| Cora | 0.6141 (±0.0599) | 0.7448 (±0.0104) | 0.8089 (±0.0078) | 0.8002 (±0.0267) | 0.7891 (±0.0269) | **0.8180** (±0.0281) | 0.8238 (±0.0109) |
| CiteSeer | 0.7253 (±0.0169) | 0.7521 (±0.0279) | 0.7876 (±0.0184) | 0.7834 (±0.0262) | 0.7884 (±0.0131) | **0.8058** (±0.0171) | 0.7951 (±0.0273) |
| PubMed | 0.8398 (±0.0044) | 0.8413 (±0.0088) | 0.8341 (±0.0097) | 0.8328 (±0.0098) | 0.8426 (±0.0128) | **0.8491** (±0.0070) | 0.8629 (±0.0017) |
| Flickr | 0.4649 (±0.0031) | 0.4868 (±0.0029) | 0.4720 (±0.0044) | 0.4844 (±0.0058) | 0.4796 (±0.0032) | **0.4937** (±0.0048) | 0.5014 (±0.0024) |
| ogbn-arxiv | 0.4495 (±0.0065) | 0.4652 (±0.0049) | 0.4685 (±0.0037) | 0.4573 (±0.0051) | 0.4761 (±0.0044) | **0.4872** (±0.0052) | 0.5386 (±0.0040) |

In the meantime, we include Fed-GNN which uses the real graph structures on the graphless clients in $\mathcal{C}_2$. Theoretically, Fed-GNN should perform better than FedGLS and the aforementioned baselines since Fed-GNN utilizes more information (i.e., graph structures on the clients in $\mathcal{C}_2$).

### 5.1.3 Parameter Settings

We implement a two-layer GCN and a two-layer MLP as the GNN model and the feature encoder in FedGLS, respectively. We apply the same model architectures to the models used in the baselines. In FedGLS, we choose a two-layer attentive encoder as the graph generator in the graph learner. The hidden size of each layer in the two models is 16. For all the aforementioned models, we use the Adam (Kingma & Ba, 2015) optimizer. The learning rates $\alpha$ and $\beta$ are set to 0.01 in the GNN model and the feature encoder and $\gamma$ is set to 0.001 in the graph learner. The temperature $\tau$ in the contrastive loss is set to 0.2. The number of local epoch $E$ is set to 5. The number of rounds is set to 100 for Cora and CiteSeer, 200 for PubMed, 300 for Flickr, and 2,000 for ogbn-arxiv. All the clients are sampled during each round.

### 5.2 Effectiveness Evaluation

In this subsection, we evaluate the performance of FedGLS over the five datasets against the baselines to answer **RQ1**. Specifically, we compare FedGLS with the five baselines on the node classification accuracy and convergence speed.

### 5.2.1 Classification Performance

We first evaluate the node classification accuracy of different approaches over the five datasets. We conduct all the experiments five times and report the average accuracy with standard deviation in Table 1. As for the baselines without preprocessing, Fed-GNNMLP outperforms Fed-MLP over all the datasets. It is because Fed-GNNMLP utilizes structure information on the clients in $\mathcal{C}_1$. However, there is still a huge performance gap between Fed-GNNMLP and FedGLS since training MLPs based solely on node features on graphless clients by Fed-GNNMLP cannot achieve comparable performance with FedGLS using structure information. Although FedProto is a personalized approach which enables each client to train a local model with global prototypes as a constraint, it does not always perform better than Fed-MLP and Fed-GNNMLP (e.g., on PubMed and Flickr). Since graphless clients in FedProto use MLPs as the backbone model and other clients use GNNs, they may learn distinct prototypes with their different backbone models. As for the baselines with preprocessing, Fed-GNNk fails to perform better than Local-GNNk on Cora and Flickr. In the meantime, we can observe significant performance degradation of Fed-GNNk compared with Fed-GNN. It indicates that constructing graph structures via kNN is not suitable for generating graphs in the real world. In the end, we observe that FedGLS consistently achieves the highest classification accuracy. Compared with Fed-GNNk,

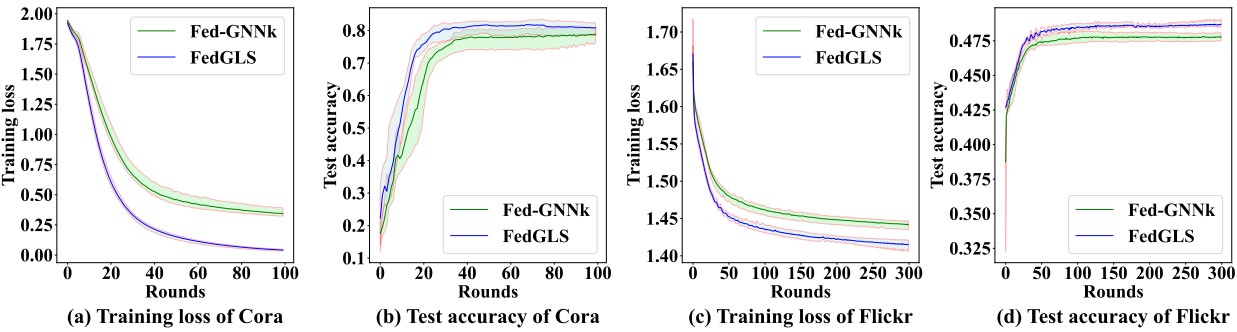

Figure 3: Result for convergence speeds of FedGLS and Fed-GNNk: (a) training loss curve and (b) test accuracy curve on Cora; (c) training loss curve and (d) test accuracy curve on Flickr.

FedGLS is able to deliver better prediction performance, much closer to Fed-GNN (even a little bit higher on CiteSeer). It is because the graph learner produces well-learned graph structures on graphless clients which benefit training of the GNN model.

### 5.2.2 Convergence Speed

We then compare convergence speeds of FedGLS with the best baseline Fed-GNNk. We present the curves of training loss and test accuracy during the training process on Cora and Flickr in Figure 3. From the results, we observe that the training loss of FedGLS decreases significantly faster than that of Fed-GNNk and the test accuracy of FedGLS reaches relatively high values with fewer rounds than Fed-GNNk on both datasets. According to the observation, we can conclude that FedGLS converges faster than Fed-GNNk. This is because the GNN model in FedGLS is trained over adaptive graphs whose adjacency matrices are generated by graph learners instead of simply produced by kNN in Fed-GNNk. The graph learners on graphless clients learn more suitable graph structures for the node classification task by learning structure knowledge transferred from other clients.

Table 2: Node classification performance (Accuracy) under different local epochs on Citeseer and PubMed dataset. Bold and underlined values indicate best and second-best mean performances, respectively.

| Datasets | Epochs | Fed-MLP | Fed-GNNk | FedGLS |
|----------|--------|---------|----------|--------|
| CiteSeer | 3 | 0.7203 | 0.7876 | **0.8017** |
| | 5 | 0.7253 | 0.7884 | **0.8058** |
| | 10 | 0.7311 | 0.7859 | **0.8015** |
| PubMed | 3 | 0.8406 | 0.8467 | **0.8477** |
| | 5 | 0.8398 | 0.8426 | **0.8491** |
| | 10 | 0.8354 | 0.8455 | **0.8493** |

### 5.3 Sensitivity Study

In this subsection, we conduct sensitivity studies to answer **RQ2**. Specifically, we evaluate the performance of FedGLS under different local epochs and graphless client ratios.

### 5.3.1 Local Epochs

The main experiments are conducted with the local epoch $E = 5$. In this part, we set the local epoch as 3 and 10 and report the average accuracy of Fed-MLP, Fed-GNNk, and FedGLS. Table 2 shows the results

Table 3: Node classification performance (Accuracy) under different graphless client ratios on PubMed and Flickr dataset. Bold and underlined values indicate best and second-best mean performances, respectively.

| Datasets | $|\mathcal{C}_1| : |\mathcal{C}_2|$ | Fed-MLP | Fed-GNNk | FedGLS |
|---|---|---|---|---|
| PubMed | 12:4 | 0.8398 | 0.8465 | **0.8527** |
|  | 8:8 | 0.8398 | 0.8426 | **0.8491** |
|  | 4:12 | 0.8398 | 0.8351 | **0.8422** |
| Flickr | 12:8 | 0.4649 | 0.4831 | **0.4982** |
|  | 10:10 | 0.4649 | 0.4796 | **0.4937** |
|  | 8:12 | 0.4649 | 0.4687 | **0.4790** |

on CiteSeer and PubMed. From the results, we observe that FedGLS performs stably under different local epochs and has the best utility under all the settings on CiteSeer and PubMed.

### 5.3.2 Graphless Client Ratios

We also consider varying ratios of graphless clients in $\mathcal{C}$. In this part, we conduct experiments on PubMed and Flickr with different graphless clients. Table 3 shows the results of Fed-MLP, Fed-GNNk, and FedGLS on PubMed and Flickr. Note that the performance of Fed-MLP keeps consistent since it does not require structure information for training. From the table, we can observe that FedGLS consistently achieves better utility compared with Fed-GNNk and Fed-MLP. In the meantime, FedGLS and Fed-GNNk show performance degradation when there are more graphless clients because of less structure information in the system. For instance, Fed-GNNk performs worse than Fed-MLP on PubMed when $|\mathcal{C}_1| : |\mathcal{C}_2| = 4 : 12$.

## 6 Conclusion

In this paper, we study a novel problem of FGL with graphless clients. To tackle this problem, we propose a principled framework FedGLS, which deploys a local graph learner on each graphless client to learn graph structures with the structure knowledge transferred from other clients. To enable structure knowledge transfer, we design a GNN model and a feature encoder in FedGLS. They retain structure knowledge together via knowledge distillation and the structure knowledge is transferred among clients during global update. Extensive experiments are conducted on five real-world datasets to show the effectiveness of the proposed algorithm FedGLS.

Although this study proposes a novel research problem in FGL, there are some limitations in the proposed FedGLS. For example, one potential limitation of FedGLS is that it may not recover the underlying graph structures on graphless clients since it produces fixed $k$ neighbors for each node on graphless clients. Therefore, the generated graph structures may not match the unknown real-world structure information. We will work on designing frameworks to obtain graph structures with consistent real-world structure information. In addition, graphless clients with heterogeneous graphs may introduce more challenges and are worthy of exploration in the future.

### Acknowledgments

This work is supported in part by the National Science Foundation under grants (IIS-2006844, IIS-2144209, IIS-2223769, IIS-2331315, CNS-2154962, BCS-2228534, and CMMI-2411248) and the Commonwealth Cyber Initiative Awards under grants (VV-1Q24-011, VV-1Q25-004).

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

## A Practical Scenarios with Graphless Clients

In this section, we provide more real-world scenarios where the proposed problem will arise.

- Graphless clients do not record edge information. In practice, some clients in an FGL system may downplay the importance of edge information and therefore choose not to record it. Considering a healthcare system with multiple hospitals, we may construct local graphs in each hospital by taking patient demographics as node features and co-staying in a ward as edges. In the early stage of a pandemic, however, some hospitals (i.e., graphless clients) may not take account of co-staying information and fail to record it. Since the co-staying information is crucial for predicting a patient's risk of contracting a contagious disease, these hospitals cannot make accurate predictions based solely on patient demographics. Instead, our proposed FedGLS can help them to generate proper graph structures with structure knowledge transferred from other hospitals, enhancing the contagious risk prediction on these hospitals.

- Graphless clients have not yet generated edge information. In the real world, some clients in an FGL system may initially have only node features, with edge information generated later. This scenario is common in dynamic graphs. Considering a financial system with multiple banks, each bank has its local dataset of customers in a city such as their demographics. For a bank operated for years in a city, it also stores transaction records among its customers and the customers naturally form a customer graph where edges represent the transaction records. Nevertheless, a new bank (i.e., a graphless client) in another city may only have few or even no transaction records and fail to form a customer graph. Considering the abundant information in transaction records, the new bank will benefit from training a model for financial lending with proper graph structures.

- Graphless clients have contaminated edge information. The edge information on some clients in an FGL system may be contaminated by malicious attackers. Considering multiple e-commerce companies that aim to jointly train a model for product rating prediction, nodes are products and edges connect products that are frequently bought together. For some companies (i.e., graphless

clients), their edge information may be destroyed or manipulated by hackers and cannot be used for model training. Without the informative edge information, it is difficult to predict the rating given to a product by its node features (e.g., word embeddings).

# B Module Designing of Graph Learner

Graph learners in FedGLS aim to learn graph structures (i.e., adjacency matrices) on the clients in $\mathcal{C}_2$ to help GCN training for node classification. Typically, a graph learner on each graphless client consists of a graph generator $\text{Gen}(\cdot)$ and a non-parametric adjacency processor $\Omega(\cdot)$. For simplicity, we omit the client index of the notations in this section.

## B.1 Graph Generator

Most existing graph generators produce a matrix $\tilde{\mathbf{S}}$ either by direct approaches (treating the elements in $\tilde{\mathbf{S}}$ as independent parameters) (Franceschi et al., 2019; Jin et al., 2020) or neural approaches (computing $\tilde{\mathbf{S}}$ through an encoder) (Chen et al., 2020b; Liu et al., 2022). Since direct approaches are difficult to train (Zhu et al., 2021), we consider neural approaches in this paper. Neural approaches take node features as input and produce matrix $\tilde{\mathbf{S}}$. Specifically, we formulate a neural network-based graph generator $\text{Gen}(\cdot)$ on each graphless client in $\mathcal{C}_2$ as

$$\tilde{\mathbf{S}} = \text{Gen}(\mathbf{X}; \omega) = \Phi(\text{Enc}(\mathbf{X}; \omega)), \tag{13}$$

where $\omega$ denotes parameters in the encoder $\text{Enc}(\cdot)$ and $\Phi(\cdot)$ is a non-parametric metric function (e.g., cosine similarity, Euclidean distance, and inner product). Here we mainly consider two specific instances of neural network-based graph generators, i.e., MLP Encoders and Attentive Encoders.

An $L$-th layer MLP Encoder employs an MLP to produce node representations by

$$\mathbf{R}^{(l)} = \text{Enc}^{(l)}(\mathbf{R}^{(l-1)}) = \sigma(\mathbf{R}^{(l-1)}\mathbf{W}^{(l)}) \tag{14}$$

for $l = 1, 2, \cdots, L$. $\mathbf{R}^{(l)}$ denote the node representations after the $l$-th layer of $\text{Enc}(\cdot)$ and $\mathbf{R}^{(0)}$ is the node feature matrix $\mathbf{X}$. $\mathbf{W}^{(l)}$ is the weight matrix of $l$-th layer in $\omega$. $\sigma(\cdot)$ is an activation function.

In an Attentive Encoder, each layer computes the Hadamard product of the input vector and parameters. Specifically, an $L$ layer Attentive Encoder on each client in $\mathcal{C}_2$ can be written as

$$\mathbf{R}^{(l)} = \text{Enc}^{(l)}(\mathbf{R}^{(l-1)}) = \sigma([\mathbf{r}_1^{(l-1)} \odot \mathbf{w}^{(l)}, \cdots, \mathbf{r}_n^{(l-1)} \odot \mathbf{w}^{(l)}]^\top) \tag{15}$$

for $l = 1, 2, \cdots, L$. $\mathbf{r}_i^{(l-1)}$ is the transpose of the $i$-th row vector in $\mathbf{R}^{(l-1)}$. $\odot$ is the Hadamard operation and $\mathbf{w}^{(l)}$ is the weight vector of the $l$-th layer in $\omega$.

## B.2 Adjacency Processor

The generated matrix $\tilde{\mathbf{S}}$ measures the similarities between the node features. With $\tilde{\mathbf{S}}$, the nodes $\mathcal{V}$ form a fully connected graph which is not only computationally expensive but also might introduce noise. Furthermore, $\tilde{\mathbf{S}}$ may have both positive and negative values while an adjacency matrix should typically be non-negative. Therefore, we deploy an adjacency processor $\Omega(\cdot)$ to refine the generated matrix $\tilde{\mathbf{S}}$ before taking it as the input of GNNs. The goal of the adjacency processor $\Omega(\cdot)$ is to obtain a sparse symmetric normalized adjacency matrix $\mathbf{S}$ with non-negative elements. Typically, the adjacency processor includes three main operations: *sparsification*, *symmetrization*, and *normalization*.

### B.2.1 Sparsification

To obtain a sparse adjacency matrix, we consider a kNN-based sparsification applied on $\tilde{\mathbf{S}}$. Specifically, the sparsification operation $\text{Sp}(\tilde{\mathbf{S}}_i; k)$ produces a sparse adjacency matrix $\mathbf{S}^{(sp)}$ by masking off (i.e., set to zero) those elements in $\tilde{\mathbf{s}}_i$ which are not in the set of top $k$ values in $\tilde{\mathbf{s}}_i$

$$\mathbf{S}_{ij}^{(sp)} = \text{Sp}(\tilde{\mathbf{s}}_i; k) = \begin{cases} \tilde{\mathbf{S}}_{ij}, & \tilde{\mathbf{S}}_{ij} \in \text{TopK}(\tilde{\mathbf{s}}_i, k) \\ 0, & \text{otherwise} \end{cases}, \tag{16}$$

Table 4: The statistics and basic information about the five datasets adopted for our experiments.

| Dataset | Clients | Node Features | Average Nodes | Average Edges | Classes |
|---------|---------|---------------|---------------|---------------|---------|
| Cora | 8 | 1,433 | 192 | 695 | 7 |
| CiteSeer | 8 | 3,703 | 149 | 535 | 6 |
| PubMed | 16 | 500 | 1,079 | 4,367 | 3 |
| Flickr | 20 | 500 | 4,441 | 28,663 | 7 |
| ogbn-arxiv | 16 | 128 | 8,948 | 50,397 | 40 |

where $\text{TopK}(\tilde{\mathbf{s}}_i, k)$ selects the highest $k$ values in $\tilde{\mathbf{s}}_i$.

### B.2.2 Symmetrization

In practice, undirected graphs typically own symmetric adjacency matrix with non-negative values. Considering this, we conduct a symmetrization operation by

$$\mathbf{S}^{(sym)} = \text{Sym}(\mathbf{S}^{(sp)}) = \frac{\sigma(\mathbf{S}^{(sp)}) + \sigma(\mathbf{S}^{(sp)})^\top}{2}, \tag{17}$$

where $\sigma$ is a nonlinear activation function.

### B.2.3 Normalization

Once we get $\mathbf{S}^{(sym)}$, we normalize it by computing its degree matrix $\mathbf{D}^{(sym)}$ and multiplying it from left and right to $(\mathbf{D}^{(sym)})^{-\frac{1}{2}}$

$$\mathbf{S} = \text{Norm}(\mathbf{S}^{(sym)}) = (\mathbf{D}^{(sym)})^{-\frac{1}{2}} \mathbf{S}^{(sym)} (\mathbf{D}^{(sym)})^{-\frac{1}{2}}. \tag{18}$$

## C   Complexity Analysis

In the federated setting, clients may not be equipped with powerful machines for model training. Therefore, the training cost becomes a major concern during collaborative training. Since FedGLS is agnostic on graph learners and graph learners are updated only once per round, we mainly focus on analyzing the computational complexity of training the GNN model and the feautre encoder (e.g., an MLP) in FedGLS. As we analyze the computational complexity during local training within a client, we omit the client index of the notations for simplicity in this subsection.

We take the training of a 2-layer GCN and a 2-layer MLP as the GNN model and the feature encoder as an example. Their parameters are trained over a graph $\mathcal{G} = (\mathcal{V}, \mathcal{E}, \mathbf{X})$ with $n = |\mathcal{V}|$ nodes on a client. Here we assume that both the GCN and the MLP have the same hidden size $m$. Typically, the GCN produces $\mathbf{Z} \in \mathbb{R}^{n \times m}$ by

$$\mathbf{Z} = \hat{\mathbf{A}} \sigma(\hat{\mathbf{A}} \mathbf{X} \mathbf{W}_1^\theta) \mathbf{W}_2^\theta, \tag{19}$$

where $\theta = \{\mathbf{W}_1^\theta, \mathbf{W}_2^\theta\}$, and $\sigma(\cdot)$ is a nonlinear activation function. The feature encoder produces $\mathbf{H} \in \mathbb{R}^{n \times m}$ by

$$\mathbf{H} = \sigma(\mathbf{X} \mathbf{W}_1^\phi) \mathbf{W}_2^\phi, \tag{20}$$

where $\phi = \{\mathbf{W}_1^\phi, \mathbf{W}_2^\phi\}$. The time complexity of the 2-layer GCN is $O(2nm^2 + 2n^2m)$ for both forward and backward pass. Hence, the overall time complexity of the GCN is $O(nm^2 + n^2m)$. Similarly, we can get the time complexity of the MLP as $O(nm^2)$. Typically, $n$ is significantly larger than $m$ (e.g., 1,079 vs 16 in PubMed). Then the time complexity of the MLP will be consequently much smaller than the GCN. As a result, the feature encoder in FedGLS does not introduce significant extra computational costs compared with other baselines such as Fed-GNNk.

To reduce the training cost of the feature encoder, we may choose to use smaller MLPs or model compression. It can not only reduce the computational complexity of the feature encoder, but also uses fewer communication resources during aggregation. We leave this exploration for future work.

## D   Datasets

In this study, we synthesize the distributed graph data on five real-world datasets, i.e., Cora (Sen et al., 2008), CiteSeer (Sen et al., 2008), PubMed (Sen et al., 2008), Flickr (Zeng et al., 2020), and ogbn-arxiv (Hu et al., 2020) by splitting each of them into multiple communities. A community is regarded as an entire graph on a client. Table 4 summarizes the statistics and basic information about the five datasets.

