# OpenReview forum: "Federated Graph Learning with Graphless Clients"
_TMLR — Accepted by TMLR_

### Review · Reviewer_K8TF · 2024-06-12

**Summary Of Contributions:**

The paper proposes a new problem formulation where some clients do not have local graph structure, and thus have to learn some approximation of it during the training. Next, the authors propose FedGLS algorithm to tackle this problem. They provide experiments comparing the proposed method with some baselines which are obtained by modifying well-known algorithms to new problem formulation.

**Audience:**

Yes

**Broader Impact Concerns:**

I do not see any ethical implications of the work.

**Claims And Evidence:**

Yes

**Requested Changes:**

Please check the strengths and weaknesses section.

**Strengths And Weaknesses:**

Strengths:
- The paper is well written and easy to read (except for some small concerns).
- The authors propose a novel problem formulation and an algorithm to solve it.
- The authors provide extensive empirical results showing the superiority against naive baselines obtained by adjusting existing algorithms to new problems.

Weaknesses:
- Not enough examples supporting that new problem formulation should be explored. In my opinion, when one proposes a new problem/field/etc. that should be explored, he/she should give many examples of where this problem appears, so that the community has a clear understanding of why this new area/field should be explored.
- In the current version, graphless clients perform more work than the clients who own graph structures. Therefore, clients who have local graph structures will stay without any jobs for some time. In my opinion, this is a waste of resources. I would suggest performing graph learning on the clients with local graph structures and forcing the output to be close to ground truth local graph structure. This knowledge can also be used by graphless clients to further improve their graph learners.

Questions:
- In the example of Figure 1, it seems new banks who don’t have transaction history are also not able to construct proper node features as well, i.e. $X^{(k)}$ seems to be unreliable as well. For example, node features should contain features like average money transferred/spent which cannot be known without transaction history. In other words, it feels like if a client doesn’t have $A^{(k)}$, then it also doesn’t have good $X^{(k)}$, and one should use node feature learner as well.
- Why is there only one step to update graph learners’ parameters while for other parts there are several local steps?
- In the introduction you mention “This scenario brings us a novel problem in the federated setting”. However, in the experiments, you mention “How does FedGLS perform compared with other state-of-the-art baselines?”. These two sentences contradict each other. Maybe it’s better to link to a section with baselines to not confuse readers.
- Why does the related work section come in the end?

---

> ### Author Response · Authors · 2024-08-01
> **Author Response to Reviewer K8TF**
>
> We appreciate the constructive suggestions from Reviewer K8TF. We hope our point-by-point clarifications can address your concerns.
>
> ---
> - W1: Not enough examples supporting that new problem formulation should be explored. In my opinion, when one proposes a new problem/field/etc. that should be explored, he/she should give many examples of where this problem appears, so that the community has a clear understanding of why this new area/field should be explored.
>
> - R1: Thanks for your constructive suggestion. We probvide another example of a healthcare system in Introduction. The red part in our modified version shows this example in detail.
>
> ---
>
> - W2: In the current version, graphless clients perform more work than the clients who own graph structures. Therefore, clients who have local graph structures will stay without any jobs for some time. In my opinion, this is a waste of resources. I would suggest performing graph learning on the clients with local graph structures and forcing the output to be close to ground truth local graph structure. This knowledge can also be used by graphless clients to further improve their graph learners.
>
> - R2: Thanks for pointing it out. The computational cost of graph learner is negligible compared with training GNN models (1 update v.s. multiple updates). Therefore, local training on normal clients will not result in apparent resource waste.
>
> ---
> - Q1: In the example of Figure 1, it seems new banks who don’t have transaction history are also not able to construct proper node features as well, i.e. $\textbf{X}^{(k)}$  seems to be unreliable as well. For example, node features should contain features like average money transferred/spent which cannot be known without transaction history. In other words, it feels like if a client doesn’t have $\textbf{A}^{(k)}$, then it also doesn’t have good $\textbf{X}^{(k)}$, and one should use node feature learner as well.
>
> - A1: Thanks for bring this up. In our assumption, node features are reliable on each client. Considering the example in Figure 1, we can use demographic information as node features. The demographics may include age, occupation, home address, credit score, and so on. We believe that these attributes are reliable to construct node features. As for unreliable node features, it may be orthogonal to our work, but it is worthy of exploration in the future.
>
> ---
>
> - Q2: Why is there only one step to update graph learners’ parameters while for other parts there are several local steps?
>
> - A2: Thanks for pointing this out. Our design of updating graph learner one time during each round can reduce computational cost of local training. We also found that one-time update is enough to obtain good performance in our empirical experiments.
>
> ---
>
> - Q3: In the introduction you mention “This scenario brings us a novel problem in the federated setting”. However, in the experiments, you mention “How does FedGLS perform compared with other state-of-the-art baselines?”. These two sentences contradict each other. Maybe it’s better to link to a section with baselines to not confuse readers.
>
> - A3: Thanks for point this out. This study proposes a novel problem in federated graph learning with graphless clients. Therefore, most existing methods cannot be applied to this setting because they require homogeneous GNN model structures across clients. However, we still find some methods, such as FedProto, compatible with this setting (although they are not specifically designed for it) and use them as baselines. We believe that Section 5.1.2 already describes this in detail.
>
> ---
>
> - Q4: Why does the related work section come in the end?
>
> - A4: We have moved Related work to Section 2.

---

> > ### Comment · Reviewer_K8TF · 2024-08-07
> > **Response**
> >
> > Thank you for responding to all raised concerns.

---

### Review · Reviewer_WwGe · 2024-07-08

**Summary Of Contributions:**

The paper proposes a new framework for federated graph learning, in which some workers (or clients, as phrased in the paper) have access to a graph and other workers don't have access to a graph—although an underlying graph exists—but have access to the graph nodes. The paper proposes an algorithm for all the workers to learn a common model to predict values for each node in such a way that the workers don't explicitly share their data.

The paper clearly explains the algorithm and also provides experiments showing the algorithm works in practice and is better than trivial extensions of previous algorithms.

**Audience:**

Yes

**Broader Impact Concerns:**

I have no concerns.

**Claims And Evidence:**

Yes

**Requested Changes:**

Some experiments are executed on only some of the datasets (for example, Table 2 considers only CiteSeer and PubMed datasets). Why is that so? Doesn't the method work well for other datasets? Some explanation is needed here (not critical for accepting the paper).

For easier readability, please add parentheses when citing papers, e.g.,
"One key problem that FL concerns is statistical heterogeneity: the data across clients are likely to be non-IID (see Li et al. 2020a)"
(not critical for accepting the paper).

**Strengths And Weaknesses:**

Strengths:
1. The proposed problem is natural.
2. The proposed algorithm is natural and interesting.
3. The experiments indicate that the algorithm works in practice.
4. The paper is well-written.

I didn't find a major weakness.

---

> ### Author Response · Authors · 2024-08-01
> **Author Response to Reviewer WwGe**
>
> We gratefully acknowledge your appreciation of our paper and hope the following clarification can address your concerns.
>
> ---
> - C1: Some experiments are executed on only some of the datasets (for example, Table 2 considers only CiteSeer and PubMed datasets). Why is that so? Doesn't the method work well for other datasets? Some explanation is needed here (not critical for accepting the paper).
>
> - A1: Thanks for pointing this out. We report the main results on all the five datasets in Table 1. As for in-depth analysis (e.g., Section 4.3.1 and Section 4.3.2), we conduct experiments on some of the datasets like other studies, e.g., FedProto, mainly due to resource constraints. Running sensitivity studies on all the five datasets may take several weeks, but we think we should get similar observations on other datasets.
> ---
> - C2: For easier readability, please add parentheses when citing papers, e.g., "One key problem that FL concerns is statistical heterogeneity: the data across clients are likely to be non-IID (see Li et al. 2020a)" (not critical for accepting the paper).
>
> - A2: Thanks for pointing this out. We modified this problem in our revised version.

---

> > ### Comment · Reviewer_WwGe · 2024-08-01
> > **Thanks for your reponse**
> >
> > Thanks for your response. I think it may help to include part of your answer A1 in the next revision to avoid confusing the reader (other readers may have similar concerns as me).

---

> > > ### Author Response · Authors · 2024-08-02
> > > **Thanks for your response**
> > >
> > > Dear Reviewer,
> > >
> > > Thanks for your reply. We will include A1 in the next revision.
> > >
> > > Best,
> > > All authors

---

### Review · Reviewer_iWP3 · 2024-07-23

**Summary Of Contributions:**

The paper introduces FedGLS, a novel framework to address the challenge of applying Graph Neural Networks (GNNs) in federated learning settings in cases where some of the clients lack of a complete graph structure. Traditional federated learning for GNNs assumes local data availability, including the graph structure for all the clients. However, in some practical scenarios this assumption can be restrictive. FedGLS overcomes this limitation by constructing local graph representations from available features and partial neighbor information, using a GNN model to capture structural dependencies and node features, and employing a feature encoder to format these features for federated aggregation. During local training, the feature encoder retains both the local graph structure knowledge and the GNN model via knowledge distillation. This way, the structure knowledge is transferred among clients during the global model update in the central node. The experimental results demonstrate that FedGLS outperforms different baselines, making it a more practical solution for federated learning with GNNs.

**Audience:**

Yes

**Broader Impact Concerns:**

N/A.

**Claims And Evidence:**

Yes

**Requested Changes:**

+ In the Introduction/motivation of the work, I think it would be nice to position the paper better with respect the horizontal vs vertical federated learning paradigms. At first, when reading these first sections it was a bit unclear to me if the method was more oriented to solve a problem for vertical or horizontal federated learning. It is a minor comment, but I think it could help to position better the paper and to make it easier for the average reader.

+ Analyze the ability of the graph learner to estimate the underlying graph structure for graphless clients.

+ Extend the discussion on the benefits, challenges, limitations, and future research in this direction.

**Strengths And Weaknesses:**

Strengths:
+ The paper introduces a novel approach for learning GNNs in federated learning settings assuming an interesting and more practical scenario where some of the clients do not have the structure of the graph. This can be of relevance for some applications, as it is the case of the financial system described in the paper as a motivating example.

+ The proposed approach to tackle the problem seems sound, including the local graph learner, the GNN model, and the feature encoder. The motivation and the steps to formulate Algorithm 1 are reasonable. The use of knowledge distillation to infer the graph structure for graphless clients is an interesting way to solve the problem.

+ Following the previous point, the paper is nicely written, the problem is well motivated, the different components of the algorithm are well presented, and the paper is easy to read.

+ The authors strived to provide a comprehensive empirical analysis using 5 different real datasets to validate their approach. The baselines used for the experiments are reasonable.


Weaknesses:
+ While the experimental results provide a nice view on the performance of the model, a deeper analysis discussing why FedGLS outperforms the other baselines would be beneficial. On the other hand, it would be interesting to analyze the performance of some of the components, like the graph learner to measure how accurate is this model in estimating the underlying (unknown) graph structure.

+ The discussion of the benefits, limitations and future work is relatively brief. It would be interesting to discuss the specific challenges encountered, the limitations of the proposed method, and further research avenues to explore in the future.

---

> ### Author Response · Authors · 2024-08-01
> **Response to Reviewer iWP3**
>
> We sincerely appreciate your efforts to review our paper and provide insightful suggestions. We hope our point-by-point clarifications can address your concerns.
>
> ---
> - W1: While the experimental results provide a nice view on the performance of the model, a deeper analysis discussing why FedGLS outperforms the other baselines would be beneficial. On the other hand, it would be interesting to analyze the performance of some of the components, like the graph learner to measure how accurate is this model in estimating the underlying (unknown) graph structure.
>
> - R1: Thanks for bringing this up. The graph learner produces well-learned graph structures on graphless clients with the structure knowledge transferred from other clients. Therefore, it can generate proper graph structures that will benefit the training of the GNN model. As for underlying graph structure estimation, we would like to clarify that the graph learner is not expected to produce graph structures consistent with the underlying (unknown) ones. Note that it aims to generate proper graph structures using structure knowledge transferred from other clients, which may be different from the underlying scenario on graphless clients. We added the above discussion in Section 6.
> ---
> - W2: The discussion of the benefits, limitations and future work is relatively brief. It would be interesting to discuss the specific challenges encountered, the limitations of the proposed method, and further research avenues to explore in the future.
>
> - R2: Thanks for your suggestion. We added more contents in purple to Conclusion.
> ---
> - C1: In the Introduction/motivation of the work, I think it would be nice to position the paper better with respect the horizontal vs vertical federated learning paradigms. At first, when reading these first sections it was a bit unclear to me if the method was more oriented to solve a problem for vertical or horizontal federated learning. It is a minor comment, but I think it could help to position better the paper and to make it easier for the average reader.
>
> - A1: Thanks for your advice. We added related contents in blue to Introduction.
> ---
> - C2: Analyze the ability of the graph learner to estimate the underlying graph structure for graphless clients.
>
> - A2: See R1.
> ---
> - C3: Extend the discussion on the benefits, challenges, limitations, and future research in this direction.
>
> - A3: Thanks for your suggestion. We added more contents in purple to Conclusion.

---

### Decision · Action_Editor_5f5P · 2024-10-02

**Recommendation:** Accept with minor revision

**Comment:**

I believe that the motivation behind the work needs to be stronger and would like to get more feedback from the authors.

**Audience:**

First, I would like to apologize to the authors for an outstandingly long review process. It is rather unfortunate that the reviewer's acted irresponsibly with their assignments but I hope that the long delay will not cause issues to the authors.
Second, the reviews and recommendations were mostly positive, but one lingering issue remains (raised by one of the reviewers and something that I had concerns about as well): what are the applications of the model? Why even involve graphless clients? Why not process their information in a more tailor-made manner for their type of data?
I will recommend that the paper be accepted with minor revisions but would advise the authors to seriously address the above issue and write a detailed explanation in the revision that will convince me to accept the paper. I unfortunately want to reserve the right to reject the paper if this explanation is not satisfactory. Since I am the only one who will read the revision (no reviewers will be involved again) I promise that the process should be much smoother.

**Claims And Evidence:**

The results are supported by clear and accurate evidence, the motivation needs improvement.

---

> ### Author Response · Authors · 2024-10-15
> **Reply to Action Editor**
>
> Dear Action Editor,
>
> Thanks a lot for your effort during the paper review process.
>
> We uploaded a revised version of our paper. In the revision, we made the following modifications to further enhance the motivation of the proposed problem.
> 1. **We replaced the original example in Introduction with a new one in healthcare (the red part).** The new example involves multiple hospitals aiming to jointly train a machine learning model for predicting whether a patient is at high risk of contracting a contagious disease. Since co-staying information (i.e., edge information) is crucial for this prediction task, training MLPs based solely on patient demographics (i.e., node features) on graphless clients will result in unsatisfactory performance. This motivates us to propose FedGLS that aims to utilize structure knowledge transferred from other clients.
>
> 2. **We added Appendix A to provide more practical scenarios where the proposed problem will arise (the purple part).** These scenarios include: 1) Graphlesss clients do not record edge information; 2) Graphlesss clients have not yet generated edge information; 3) Graphlesss clients have contaminated edge information. We provide real-world applications from various domains, including healthcare, finance, and e-commerce for these scenarios. These scenarios indicate that our porposed problem is important in practice and morivate us to design FedGLS to handle it.
>
> 3. **We emphasized the performance degradation of Fed-GNNMLP in Section 5.2.1 (the blue part).** Fed-GNNMLP enables normal clients to collaboratively train a GNN model while graphless clients joinlty train an MLP model. In this way, graphless clients will completely ignore potential graph structures. According to the results, Fed-GNNMLP suffers from significant performance degradation compared with our method.
>
> We hope the above modifications can fully address your concerns about the motivation of our paper. We will be willing to provide more explanations if you have any further questions.
>
> Best,
> Submission 2595 Authors